# Astrocytes: News about Brain Health and Diseases

**DOI:** 10.3390/biomedicines8100394

**Published:** 2020-10-06

**Authors:** Jacopo Meldolesi

**Affiliations:** Department of Neuroscience, San Raffaele Institute and Vita-Salute San Raffaele University, via Olgettina 58, 20132 Milan, Italy; meldolesi.jacopo@hsr.it; Tel.: +39-02-2643-2770

**Keywords:** astrocyte heterogeneity, immunofluorescence, traumatic injury, astrocyte senescence, tau and tauopathies, AD, Alzheimer’s disease, SASP, senescence-associated secretory phenotype, ARTAG, aging-related tau astrogliopathy

## Abstract

Astrocytes, the most numerous glial cells in the brains of humans and other mammalian animals, have been studied since their discovery over 100 years ago. For many decades, however, astrocytes were believed to operate as a glue, providing only mechanical and metabolic support to adjacent neurons. Starting from a “revolution” initiated about 25 years ago, numerous astrocyte functions have been reconsidered, some previously unknown, others attributed to neurons or other cell types. The knowledge of astrocytes has been continuously growing during the last few years. Based on these considerations, in the present review, different from single or general overviews, focused on six astrocyte functions, chosen due in their relevance in both brain physiology and pathology. Astrocytes, previously believed to be homogeneous, are now recognized to be heterogeneous, composed by types distinct in structure, distribution, and function; their cooperation with microglia is known to govern local neuroinflammation and brain restoration upon traumatic injuries; and astrocyte senescence is relevant for the development of both health and diseases. Knowledge regarding the role of astrocytes in tauopathies and Alzheimer’s disease has grow considerably. The multiple properties emphasized here, relevant for the present state of astrocytes, will be further developed by ongoing and future studies.

## 1. Introduction

The discovery of astrocytes, the most numerous glial cells in the brain, occurred as soon as the study of brain cytology started. Astrocytes, different with respect to neurons, were already recognized in the exciting images produced by Cajal at the end of the nineteen century. At the time, however, for many decades, the complex of astrocytes was described as a glue, a continuous and structured multicellular complex providing adjacent neurons with molecular exchange and metabolic support. Only 25–30 years ago, an astrocyte “revolution” started with the recognition of various functional properties typical of these cells. Additional properties were identified and characterized by subsequent studies. In other words, astrocytes were recognized to operate in discrete territories of the brain tissue where they interact with specialized regions of the neuronal plasma membrane: at the cell body, the axon, and at hundreds of synapses [1,2].

The secretory activity of astrocytes, called gliotransmission, occurs by exocytosis, with various transmitters released by vesicles at synapses [1,3]. Two such transmitters (glutamate and ATP) are the same as those of neurons. Others (D-serine and eicosanoids) are more specific. Additional, highly active and trophic factors are also released by exocytosis. In addition, all mentioned factors are released also via channels of the plasma membrane. The released factors activate receptors expressed by neurons, various types of glia, and blood vessels. Within astrocytes and other cell targets, these factors induce transient increases and/or oscillations of cytosolic Ca^2+^; however, they are slower however with respect to those of neurons [4,5]. The feed-back and feed-forward signaling effects induced by astrocytes can ultimately tune the balance of neurons between excitation and inhibition [6]. In physiological conditions, astrocytes contribute to the state of the target cells, up to the protection of neurons.

The study of brain diseases has clarified the participation of astrocytes in various processes, from physiological events to strokes and neurodegenerative diseases, including the Alzheimer’s disease (AD). The role of astrocytes is known to vary during aging, with its contributions extending from the maintenance of health conditions to the development of diseases [7,8]. Aging, in fact, does not preclude the ability of astrocytes to maintain a positive environment in the central nervous system. The state of neurons depends on their interaction with neighboring cells and also on the inflammation often established in the brain. Some of these properties have already been presented in coordinated reviews, a few of which including recent presentations of diseases [9,10].

My interest in astrocytes started from the discovery of their classical properties and functions partially summarized in the previous paragraphs. This initial interest was strengthened over the last few years, when intense studies have discovered several new properties, functions, and defects of astrocytes, which have attracted more and more interest also in the scientific community. At present, therefore, the “revolution” of astrocytes, developed before the 2000s [1], has been expanded in many directions. Many properties and functions typical of neurons or other glial cells have not been attributed to astrocytes. In other words, most properties and functions of astrocytes have remained the same; however, their relevance has become increasingly detailed and important.

The increased importance of astrocytes is generally recognized. Therefore, reviews written by others were conceived either as comprehensive reviews or focused on a single function. In view of my interest for the role of astrocytes in several directions, I decided to write my review according to a different strategy, focused on six areas, in which recent studies are presented separately: heterogeneity, inflammatory responses, strokes and repairs, senescence, tau and tauopathies, and AD. To provide information regarding the previous knowledge, each recent area is preceded by short presentations of relevant data published several years ago. To emphasize that the six areas are not fully separate from each other, the conclusion includes a few short comprehensive considerations that illustrate important findings obtained in various areas that are interactive with each other.

## 2. The Heterogeneity of Astrocytes

In the past, the structural homology of astrocytes was a common opinion. Such a property was considered critical, on one side, for the general and prolonged interaction of these cells with the vessels of the blood-brain-barrier (BBB); on the other side, for the connection of their branches to various neuronal components: bodies, axons, and thousands of synapses. At that time, astrocyte heterogeneity was limited to resting and stimulated conditions. Around 2010, however, astrocyte heterogeneity began to be recognized among brain regions and within single regions. These differences were due to gene dependence as well as to stimulation and disease responses [11]. At present, the research is clear that the properties of astrocytes, widespread in young age, are also maintained in mature mice and rats if in good health [12,13,14,15]. During aging, however, many changes and alterations have been reported [16,17,18,19]. The interactions of heterogeneous astrocytes with neurons are critical for many key events of brain development and function, including the formation and function of synapses, their release and reuptake of transmitters, the production of trophic and toxic factors, and the control of neuronal survival [12,13] (Figure 1). These interactions are due to the molecular properties of astrocytes, with differences in the expression of distinct genes (a) at various stages of their functions; (b) at various regions of the brain [20]; (c) in neuronal circuit regulation [21], and (d) also in the course of their pathologies [16,17,18,19]. 

As summarized in [22], six different forms have been recognized in different diseases of astrocytes. This is also specified in our sixth section, Tau and Tauopathies: plaques, tufted, ramified, and globular astrocytes predominate in primary tauopathies and torn-shaped and granular fuzzy astrocytes predominate in aging-related tau (ARTAG) tauopathies (Figure 1). Many region-specific transcriptional properties change in a region-dependent manner. The traditional effects of astrocyte aging, induced by their interaction with neurons, vary depending on the state of the latter cells, healthy or sick. Changes are accompanied by forms of neuroinflammation sustained by cytokines, released by astrocytes and microglia [19]. In many such cases, the astrocyte interactions/communications with neighboring cells are reduced, while their expression of relevant factors, including brain-derived neural factor (BDNF) declines progressively [19,20,23].

## 3. Neuroinflammation

Neuroinflammation is a process traditionally attributed to cells other than astrocytes, such as microglia, lymphocytes, and macrophages. During the last decade, however, the role of astrocytes has been reconsidered and found to be more important than previously believed [24]. For example, a role of astrocytes was found to be necessary for the activation of neuroinflammation by T lymphocytes [25]. Upon exposure to mesenchymal cells, astrocytes were induced toward the expression of pro-inflammatory genes and to the release of their cytokines [26,27].

At present, neuroinflammation is known as a process widely expressed by astrocytes. The intensity varies in most aspects of brain life, from development to damage, and from regeneration to repair. Excesses of at least some of these conditions can be interpreted as hallmarks of various diseases, including neuronal neurodegeneration. In addition to other pathological conditions, such as stroke, trauma, genetic diseases, and para-neoplastic disorders. Neuroinflammation can also be activated during immune responses [28,29]. In these conditions, astrocytes do not operate alone. To activate most pro-inflammatory conditions, they require the co-operation of microglia [30,31,32]. In neurodegenerative diseases, processes reducing the inflammatory conditions reduce the pathology, and thus promote healthier lives of old patients.

In recent years, the stimulation of neuroinflammation sustained by cooperation with microglia has been shown to depend on the conversion of astrocytes to their reactive version A1 [14]. In old rats, the physical interaction of three types of brain cells (astrocytes, microglia, and neurons) induced the appearance of an aggressive mechanism that was dependent on pro-inflammatory cytokines, including tumor necrosis factor-α (TNF-α) and interleukin-1β (IL-1β). Such responses, which are absent or weak in young rats, increase progressively in aging animals, with the fragmentation of neurons and ensuing clearing of their debris. With time, the survival of neurons and synapses decreases progressively [31]. The combination of low A1 astrocyte with leptin, an adipose tissue-derived hormone, has been shown to strengthen the astrocyte responses, especially in the hypothalamus, induced by activated microglia and sustained by increased TF-kB gene transcription [23,33]. Other endocrine agents can, in contrast, reduce the responses. This is the case of menin, a protein that, in aging patients, reduces neuroinflammation by reducing the release of IL-1β [34].

In all these cases, opposite responses are sustained by the activation or reduction of astrocyte effects [34]. The mechanisms of neurodegenerative responses by astrocytes do not depend only on cytokines and hormones. They can be modulated by additional factors and activated by cell processes. This is the case of the calcineurin/NFAT-Ca^2+^ signaling, with ensuing synapse dysfunction and glutamate excitotoxicity, activated during various diseases and by healthy agents [35]. The tuning of neurodegenerative responses is induced in astrocytes by autophagy. Such a process, via the activation of lysosomes, reinforces the degradation of cytoplasmic proteins together with the oxidative damage and neurodegenerative disorders of aged cells [36]. In conclusion, inflammatory responses, important for astrocytes, are critically relevant during aging, as demonstrated by the direct comparison of healthy and diseased animals and men.

## 4. Traumatic Injuries

Traumatic lesions of the central nervous system, most frequently brain strokes and spinal cord injuries, are frequent diseases of human life. Astrocytes are the only brain cells that, in parallel to potentially detrimental effects, play functions of great importance for neuron recovery. In the case of ischemia followed by re-perfusions, astrocytes not only survive but increase in number, with the significant protection of neurons [37]. Beginning 10 years ago, appropriate manipulations of astrocyte functions were expected to contribute to the development of important strategies, efficient for neuronal survival [38], often requiring axonal repairs of interest [39]. Therefore, a traumatic injury section is included in the present review.

The first interest upon injury is the neuro-genetic restoration of lost neurons. From in vitro studies, the potential contributions of astrocytes for such attempts appear almost unlimited. Extensive in vivo studies have shown that, upon focal ischemic injury, astrocytes re-established the equilibrium that was affected. For such a process to take place, astrocytes must be epigenetically reprogrammed to become neurons. In rats, positive results were obtained during the last five years involving miRNAs and growth factors [40,41,42]. Details regarding the processes in which post-mitotic astrocytes, starting from inflammatory signals, replace other cells, such as stem cells, are beyond the present discussion. However, a whole area of astrocyte involvement is expected to develop in the near future [43,44].

The relevance of astrocyte reactions to traumatic brain injury is considerable for its therapeutic potential [45,46,47]. Details regarding the roles of astrocytes in various conditions were clarified. First, the results obtained after ischemic strokes in neonatal or young animals were profoundly different from those obtained in adult and aged animals. The lesions of adult/old animals were more severe at various levels for the BBB, inflammation, and reactive oxygen species [48]. A parallel study in rats, exposed to acute traumatic brain injury, found progressive differences in populations with age from 4 to 18 months. In the latter, the lesions were considerable and persistent, whereas in young animals they were smaller and transient. Parallel investigation of the gene expression also revealed considerable differences that were much more severe in adults [49]. 

In other experiments, differences were found in the astrocytes of animals exposed to injury with or without pre-treatment with the transforming growth factor β1 (TGF-β1), a regulator of cell survival and differentiation [50]. Other experiments on spinal cord motor neurons were carried out in old rats transplanted with human neural progenitor cells, which differentiate into astrocytes undergoing maturation. The results demonstrated that astrocytes protected the aging motor neurons against loss [51]. Analogous effects were obtained by inducing an astrocyte miRNA, miR-21. Upon the activation of TGF-β1, miR-21 is known to induce profound positive changes in astrocyte responses, mediated by an important signaling pathway [52]. In contrast, the activation of another miRNA, miR-335-3p, was found to increase the severity of negative astrocyte responses by changing the properties of their metabolic effects [53]. All together, these results confirm that, in the central nervous system, astrocytes are the only cells competent for the reduction and repair of traumatic lesions. However, astrocytes are very efficient when the brain is young and become progressively inefficient during aging.

## 5. Astrocyte Senescence

Senescence is a process that occurs in most types of cells. Toward the end of last century, studies carried out in some cell types demonstrated that senescent cells, shortly after their establishment, undergo an irreversible arrest of their cycle. Upon subsequent events governed by the senescence machinery, the cells reach the chronological events planned for their age advancement, up to the end of their life. In astrocytes and microglia—the two types of glial cells that exhibit an altered pro-inflammatory profile—brain senescence begins earlier [54]. Their senescence-associated secretory phenotype (SASP) releases IL-6 and IL-8 together with extracellular proteins, such as matrix metalloproteinase-3 (MMP3) and -10 (MMP10), and TIM2. These astrocytes favor the development of further inflammation and chronic neurodegeneration. In addition, senescence stimulates the release of splicing factors and, thus, acquires the complexity of their gene expression [55,56] (Figure 2). By a comprehensive pharmacological approach, SASP release was shown to be regulated by a Nuclear Factor-κB (NF-kB) pathway [57,58]. By these developments, the phenotype typical of senescence is converted, including some properties of neurodegeneration, typical of peculiar diseases [59].

Astrocyte senescence has been investigated also in terms of the transcriptome. Several genes indicative of astrocyte responses can be down-regulated, and pro-inflammatory genes are up-regulated. The function of senescent astrocytes appears, therefore, to meet the pathogenesis of brain disorders [60] (Figure 2). Among the essential alterations induced by senescence in astrocytes are the downregulation of K^+^ and the glutamate transporter, with the ensuing decrease of neuronal protection. These events, and the ensuing excitotoxicity, appear as events that favor the development of Alzheimer’s and other dementias [57,58,59,60]. Due to these properties, senescent astrocytes appear as potential targets, interesting for the development of innovative therapeutic treatments [57,58,61].

## 6. Tau and Tauopathies

Hyper-phosphorylated tau is known as a critical component of AD. In addition, tau is known as a key factor in other diseases, such as tauopathies and glial and neuronal pathologies, and is heterogeneous in various respects, which share many important properties of dementia [62,63,64]. For many years, tauopathies were considered of neuronal origin; however, glial cells, astrocytes, and oligodendrocytes, are also involved. In this case, the question was whether tau is generated also in glial cells or is only of neuronal origin [65,66,67]. The function of astrocytic tau and tauopathies has increasingly been considered independent on neurons and AD [64,65,66]. In other words, the origin of tau in astrocytes and oligodendrocytes has been investigated directly [66,67,68]. The possibility of exclusive neuronal origin was turned down. In a mice model where tau expression by neurons had been knocked-out, the expression in glial cells was maintained [69]. Interestingly, tau expression by both glial cells was not followed by intense spreading to other cells. Thus, in the brain, tau is generated in parallel by neurons and both glial cells [69] (Figure 3).

Knowledge regarding tau in astrocytes has been derived from multiple pathologies. As already mentioned in the Section 2, The Heterogeneity of Astrocytes, six types of astrocytes, distinguished by their morphology [22], differ in their disease and intracellular distribution. The plaques, tufted, and ramified astrocytes, together with globular inclusions (GGT), predominate in a form of diseases, the primary gliopathies and their proteins are defined as protein astrogliopathies. The thorn-shaped and granular/fuzzy astrocytes predominate in the disease called ARTAG. The latter is distributed primarily in white and gray matter, in subpial and subependimal locations and at the perivascular BBB (Figure 1). Among ARTAG properties are the tau sites of their phosphorylation, partially different from those of neurons, combined with interactions with other phosphorylated proteins [70,71,72,73]. On the other hand, a fraction of ARTAG astrocytes, associated with the primary progressive aphasia phenotype of the AD cohort, correlates with older ages [73,74], while primary astrocytes are distributed more widely than ARTAG astrocytes. Among their functions is the conversion of novel insights into brain aging and neurodegenerative diseases [70,71].

An interesting aspect of tau in astrocytes depends on a transcription factor, transcription factor EB (TFEB). By the expression of specific genes, this factor governs the activity of lysosomes. An initial study demonstrated this TFEB to increase the uptake of Aβ into astrocytes followed by its digestion within lysosomes, inducing decreased pathogenesis of AD [75]. Recently, the increased uptake of Aβ, induced by TFEB and followed by lysosomal digestion, were found to affect the extracellular tau, with a positive effect on ARTAG and various tauopathic diseases [76,77].

## 7. The Role of Astrocytes in Alzheimer’s Disease

Astrocytes play critical roles in various neurodegenerative diseases. Here, I will focus only on AD, the most important and most intensely investigated disease, leaving out the others. For decades, neurons were considered the cells playing the major role in the development and severity of AD, the alterations, and ensuing death. Astrocytes were known to be present, mostly surrounding the amyloid plaques; however, until about 10 years ago, their role and their interactions with neurons remained uncharacterized and poorly understood [78]. During the last few years, critical forms of the disease became more and more dependent on factors released by glial cells. Upon such insights, the focus of research, away from neurons, has largely moved toward glial cells, primarily astrocytes [79,80,81,82]. Consequently, knowledge regarding the role of astrocytes in AD has grown considerably. Here, I will report about the various developments now established.

A critical process sustained by astrocytes at various stages of AD is their reactivity. The latter process, established by astrocytes in response to various actions, results in the inhibition of neuronal resting activities, such as regulated homeostasis, synaptic plasticity, and neuroprotection. Concomitantly, astrocytes, structured in A1-like conditions [14], grow up in volume. For these developments, astrocytes are always complex and heterogeneous. Their identification is often based on the recognition of various markers, which are present only in a fraction of their population. This observation may explain why the effects of astrocytes in aging patients are variable. In many cases, astrocytes acquire aggressive properties, with a gain of toxic functions and loss of neurotrophic effects [83,84]. In other cases, however, astrocyte reactivity induces metabolic changes, including the JAK2-Stat3 cascade, contributing to reductions of amyloid deposition and restorations of defective synapses [85,86,87]. Thus, the final effects largely depend on the general state of participating astrocytes.

Another important process by astrocytes in the development of AD is the specific secretion of relevant agents. Apolipoprotein E (ApoE, active especially as ApoE4), carries lipids to neurons, plays an undisputed role in AD pathology. The secretion of this protein from astrocytes is positively controlled by Axl, a tyrosine kinase receptor. This mechanism supports the development of AD, while inhibition of the receptor results in the inhibition of ApoE release and, thus, in a slowing down of the AD process [88].

Analogous effects are induced by the protein phosphatase 2A (CIP2A), abundant in several cancer cells, which is also expressed by astrocytes. This expression stimulates the release of cytokines and the accumulation of Aβ, with the induction of synaptic degeneration and additional symptoms [89]. However, the astrocyte secretion of other agents, such as the protein thrombospondin-1 and the TGF-β1, can induce a reduction of various steps of AD pathology [90,91].

Recent studies revealed that, in AD and Parkinson’s, events reinforced by astrocytes are sustained by various processes of cell biology. The phagocytosis of apoptotic cell debris can induce various effects. In some cases, neuronal material is accumulated within lysosomes, with ensuing defects of the organelle with the extracellular release of toxic debris [92]. In other conditions, the uptake of Aβ within astrocytes results in digestion and, thus, in the control of its level [93]. An important event in AD is the progressive deletion of excitatory amino acid transporter 2 (EAAT2), a major glutamate transporter abundant in astrocytes but present also in neurons [94]. The lack of the same transporter in the two types of cells yields different defects: in astrocytes, an early defect of short-term memory, special reference learning, and also long-term memory; in neurons, of only long-term memory [94]. Neurons and astrocytes have been known for years to exchange extracellular vesicles [95]. The exchange effects induced by such types of intercellular communication can be variable [96]. In aged AD patients, most exosomes, released by reactive A1 astrocytes, activate various neurotoxic mechanisms, inducing fragmentation and death of neurons [96,97].

Interesting studies of the last few years have revealed the young precocity of long-term AD, whose patients begin expressing molecular alterations years before the appearance of clear symptoms. Among the relevant early data observed are the decreased astrocyte proliferation, the decreased expression of enzymes, such as glutamine synthetase, and the expression of a population of genes. Highly interesting are the RNA sequences, known to identify the disease-associated astrocytes. The latter properties, present already at early levels, increase with the progression of the disease [82,89,98].

The majority of the recent astrocyte data regarding AD were obtained by studies of mouse brain models [84,85,87,89,94]. Additional data were obtained from mouse astrocyte lines, in some cases, duplicated with human lines or cultures [78,88,92]. Thus, the validity of the data for the human disease has not yet been finally demonstrated. This problem is relevant as rodent astrocytes differ considerably and in many respects from human astrocytes, including the functionalities and gene expression, the recent development of research, including the use of human astrocytes derived from stem cells, and the analyses of single nucleus RNA sequencing, appear of interest for future in vivo studies, focused on the re-consideration of multiple aspects of the disease and its perspectives [78,82].

## 8. Conclusions

Compared to many other reviews on astrocytes, this presentation can be considered innovative. My strategy was neither comprehensive nor focused on a single area. Rather, it includes the distinct presentation of six areas intensely developed during the last few years. The six areas I chose are not the only ones with recent astrocyte development. For example, an important function previously attributed primarily to neurons—the circadian regulation governed by the brain clock—has been recently recognized as related to astrocytes [99,100]. However, circadian regulation is not easy to coordinate with other brain functions. Therefore, it was not chosen. Similar problems occurred with a few other activities. Thus, our choices remained limited to six.

As already mentioned in the introduction, a common property of the six astrocytic areas presented separately here, is their functional coordination within the cells. This is the case of astrocyte heterogeneity, useful for the development of inflammation, the restoration of structural lesions, and the distinction between aging of healthy humans from AD and tauopathy patients. Neuroinflammation is relevant for its unexpected participation in the discussed pathologies [24,28]. For some traumatic injuries, astrocyte cooperation exists with other areas: heterogeneity, inflammation, and neuronal diseases [37,39]. Senescence appears useful to support the dual role of astrocytes during aging, in healthy humans and diseased patients, protecting neurons or contributing to the toxicity [56,57]. 

Tau is an important component of AD and the key factor of tauopathies. In healthy humans, tau is also present, although at lower levels [69,72]. Finally, AD initial symptoms, which sometimes involve astrocytes, are present in young, apparently healthy humans, destined to become patients with aging. The astrocyte presentation of the disease is focused on the coordination with the other areas and in the development of new pathogenesis [89,90]. The relevance expected for future developments includes a general reconsideration based on the role of astrocytes, new innovative approaches to the understanding, and new therapies of diseases [46,47,82,98].

As emphasized in our introduction, for many decades, the functional role attributed to astrocytes in the brain remained, at best, marginal. Innovative progress led to a recognized and appreciated “revolution” [1], growing over the following three decades. This review illustrated the exciting functional improvements developed during the last few years in six areas, including the most important in the brain. In addition to the present state, I am impressed by the progress expected in the near future, particularly for the missing knowledge, symptoms, and therapy of diseases, such as the tauopathies and AD discussed here. In conclusion, except for the special function types of other brain cells, astrocytes have become, or appear to be becoming, the most important cells for many relevant brain functions.

## Figures and Tables

**Figure 1 biomedicines-08-00394-f001:**
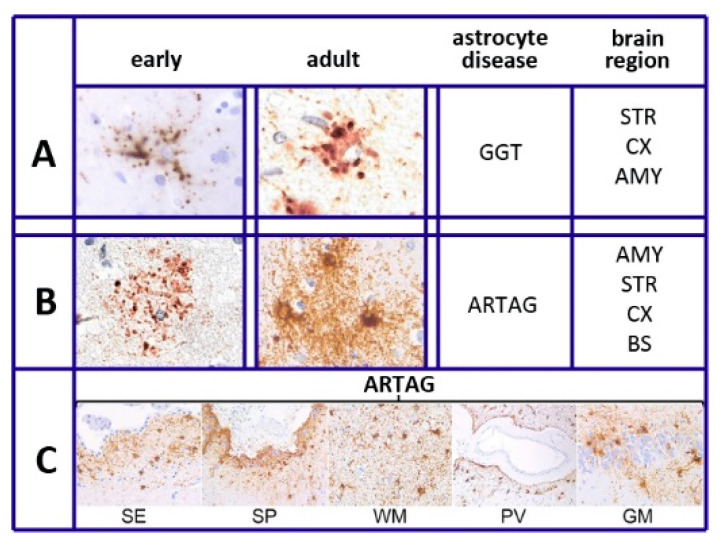
Tau heterogeneity and pathology of various astrocytes. (**A**,**B**) show the tau deposition of globular-glial tauopathies (GGT) and aging-related tau astrogliopathy (ARTAG), presented as early (left columns) and adult (right columns) morphologies. Brain regions of GGT and ARTAG distribution are listed on the right: AMY = amygdala; STR = striatum; CX = cortex; BS = brainstem. (**C**): thorn-shaped astrocytes of ARTAG type distributed at subependymal (SE), subpial (SP), white matter (WM), perivascular (PV), and rare gray matter (GM) location. Reproduced in part with permission from Figure 1 of [22]. Thanks to G.G. Kovacs and the Frontiers Production Office.

**Figure 2 biomedicines-08-00394-f002:**
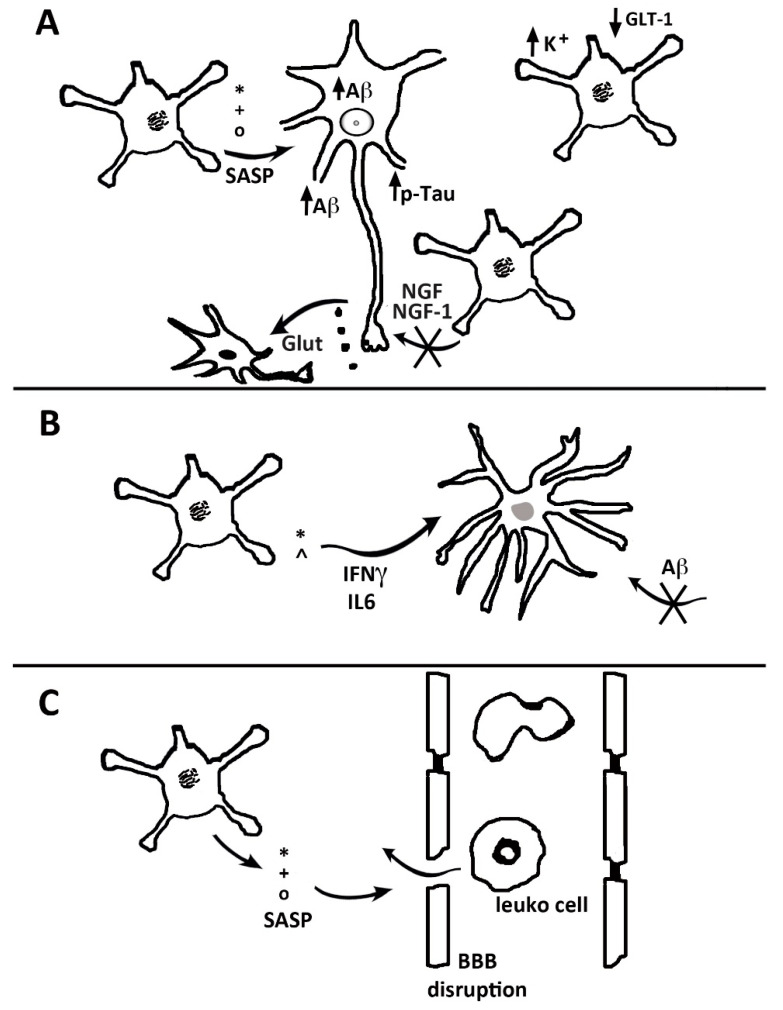
Role of senescent astrocytes in the neuronal, microglial and vascular pathologies. In (**A**), senescence-associated secretory phenotype (SASP) factors (*, +, °), released by senescent astrocytes (striped nuclei), increase in neurons the generation and deposition of Aβ together with the phosphorylation of tau. Neurons release glutamate (black dots, Glut), which cannot be cleared from the synaptic environment by senescent astrocytes, leading to an accumulation of extracellular excitotoxic glutamate. Together with the reduced secretion by senescent astrocytes of neurotrophic and growth factors (NGF and IGF-1), the excess glutamate may lead to decreased neuronal growth and/or increased neuronal death (black nucleus). In (**B**), a senescent astrocyte secretes IFNγ and IL-6, which activate microglia (gray nucleus) and reduce the Aβ uptake. In (**C**), SASP factors disrupt the endothelial tight junctions, inducing local BBB disruption with leukocyte transmigration.

**Figure 3 biomedicines-08-00394-f003:**
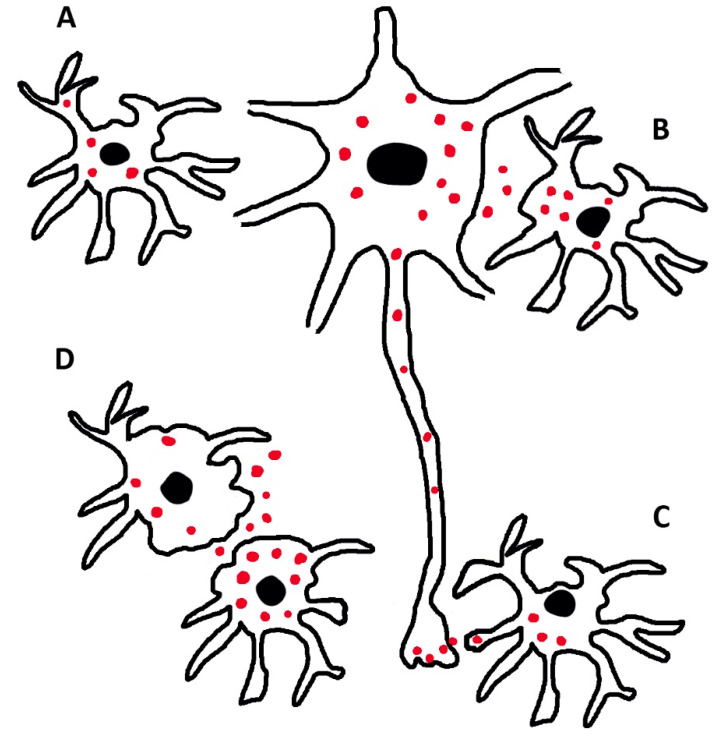
Distribution and dynamics of phosphorylated tau in astrocytes and neurons. In (**A**), an astrocyte contains intracellular tau (red dots) possibly generated locally. In B and C, a neuron interacts directly with astrocytes, exchanging tau at its body (**B**) and its synaptic (**C**) level. (**D**) shows two astrocytes with tau distributed in their cytosol and in their extracellular space. The interaction of these glial cells with a neuron is not excluded. This could occur primarily at a distance.

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
