# Peer review of "Astrocytes: News about Brain Health and Diseases"

_biomedicines, 2020, doi:10.3390/biomedicines8100394_

Round 1
Reviewer 1 Report
This is very interesting review, I have only minor comments:
1) Abstract: The author talks about six function, it would be mention the reason of their selection in the abstract.
2) Line 41: please chnage the word addressed by some more appropriate.
3) line 66: please specify, in what were studies successful.
4) line 92: tau pathology Section: please consider better reference.
5) In general: I suggest to change the chapter numbering: 1. Introduction, 2. Heterogeneity of astrocytes, 3. Areas of recent studies (or better Overall title): 3.1 Neuroinflamation, 3.2 Traumatic injuries and so on....4. Conclusion
Author Response
I thank very much Reviewer 1 for the positive comments about my review on Astrocytes. I have taken into consideration all the criticisms and suggestions. Specifically
- I have added in the Abstract an explanation about the six properties of astrocytes recently discovered and investigated in this review.
- I have changed the sentence of line 41, as recommended.
- line 66. I have reconsidered the results presented, specifying the more successful studies, as suggested.
- line 92. I have reconsidered the initial paragraph of Section 2, that dealing with heterogeneity, reconsidering specific aspects and reorganizing the order of references.
- Concerning the change in the organization of the review, I have thought and decided that your suggestion was incompatible with the strategy of my work. Discovery of the heterogeneity of astrocytes has opened chances to Sections of other content, as emphasized in the final Conclusions of this review. I think that heterogeneity covers an important aspect of the review possibly integrated with some others, and can therefore be accepted as established in the original presentation of the review.
Reviewer 2 Report
The present review article was not so novel and innovative but interesting to demonstrate numerous astrocyte functions reported in recent years.
Because there were some errors, minor spell check was required.
Author Response
Reviewer 2 was positively impressed by the present review she/he considers not very innovative but accurate in its presentation. She/he recommended a revision of the language of the text, which has been done.